# Tunnel and underground engineering rock mass water inrush damage and acoustic emission characteristics

Jiajun Zeng[1]*, Chengzhi Pu[2], Qiyun Wang[1], Qingqing Shen[1], Qiang Zeng[1], Zhicheng Yang[1]

**1** College of Civil and Construction Engineering, Hunan Institute of Technology, Hengyang, China, **2** School of Resource Environment and Safety Engineering, University of South China, Hengyang, China

\* Zengjiajun93@163.com

**Data Availability Statement:** All relevant data are within the manuscript and its Supporting Information files.

## Abstract

To achieve the actual situation of water pressure stabilization during underground and tunnel water inrush disasters, the team independently developed a stable water pressure test system and conducted fracture and failure tests on fissured rock masses under the coupling effect of 1MPa stable water pressure and stress and without water pressure. Combined with data collected by acoustic emission instruments, the mechanical characteristics of fracture and failure, crack propagation mechanism, and acoustic emission response mechanism of fissured rock masses under the coupling effect of stable hydraulic pressure and stress were studied. The results showed that throughout the entire experimental process, the hydraulic pressure remained continuously stable, with a decrease of only 0.14%; The variation pattern of peak strength of fissured rock mass with increasing crack inclination angle under stable hydraulic pressure changes from a decrease and then an increase in the absence of hydraulic pressure to an increasing trend; The crack propagation length of low angle fissured rock mass is generally higher than that of high angle fissured specimens. The longer propagation path increases the range and effect of hydraulic pressure, and the initial crack propagation length of fissured rock mass under hydraulic pressure is also significantly longer than that of specimens without hydraulic pressure; During the loading process, both the acoustic emission ringing count and damage variable can be divided into four stages. From the cumulative total number of acoustic emission ringing counts, it can be seen that during the loading process, the total number of acoustic emission ringing in fissured rock masses subjected to hydraulic pressure is significantly lower than that of specimens without hydraulic pressure, and the trend is also relatively stable.

## 1 Introduction

The hydrogeology and structure forms of the tunnel and underground geotechnical engineering are quite complicated. Water and mud inrush accidents are common, harmful, and difficult to control [1]. Water-rich, high-pressure, and bad geology are the main geological conditions that induce water inrush and mud outburst disasters. Construction disturbance is

**Funding:** This work is supported by Hunan Provincial Natural Science Foundation of China (2023JJ50104); Scientific Research Fund of Hunan Provincial Education Department (21B0803, 22B0853, 23B0838); Hunan Institute of Technology provincial-level applied characteristic discipline (KFB23022); National College Student Innovation and Entrepreneurship Training Program Project(S202311528042). Those funders play important roles in the study design, data collection and analysis.

**Competing interests:** The authors have declared that no competing interests exist.

the external cause of the accident. Under the joint action of hydraulic pressure and engineering disturbance, the surrounding rock medium generates fracture behavior, forming the channel of water inrush and mud outburst, leading to the accident of water inrush and mud outburst, which seriously affects the stability of the surrounding rock. Water inrush is often characterized by uncertain orientation, large water inrush, strong hydraulic pressure, slow water inrush decay, long duration, etc., which is very difficult and harmful to prevent and control [2]. Therefore, it is of great significance to carry out experimental research on mechanical properties and deformation rules of rock mass under the action of hydraulic pressure for effective prevention of water and mud inrush disasters in tunnels and underground engineering.

When encountering fault-fractured zones during mining and tunnel excavation, the granular structure of the fractured surrounding rock is prone to forming water channels, which can easily cause water inrush hazards [3–5]. In order to study the influence of the interaction between rock fractures and water on the fracture and failure mechanism of rock masses, Li [6] conducted hydraulic tests on sandstone specimens containing single fissures. Zhou Zhihua [7] prepared specimens with internal fissures using paper instead of fissures, and conducted uniaxial and cyclic loading and unloading tests on the internal fissure rock mass under the action of osmotic water pressure; Guo Kongling [8], Mei Jie et al. [9] prefabricated elliptical three-dimensional fissured rock specimens and studied the fracture mode of fissured rock mass under the influence of hydraulic pressure. Fu Jin wei [10], Yang Lei [11], and others used transparent resin materials to prepare transparent resin specimens with built-in three-dimensional cracks. They conducted uniaxial compression fracture experiments on crack specimens under the influence of crack hydraulic pressure and studied the initiation and propagation mode of newly formed cracks near the prefabricated crack surface under the influence of crack hydraulic pressure. Cao Ping [12] and Hao Rui qing [13] prepared fissured yellow sandstone specimens, sealed the hydraulic pressure with epoxy resin material, and studied the peak failure strength law and crack propagation and failure mode of fissured rock masses. Wei Chao [14], Li Yong [15], and others designed a water-sealing fixture, using sealing washers and organic glass plates to encapsulate penetrating cracks through hydraulic pressure. They analyzed the influence of hydraulic pressure on the fracture mechanical characteristics of crack specimens and crack propagation and failure modes.

Acoustic emission testing technology can effectively monitor and reflect the evolution and propagation process of internal cracks in materials and is widely used to study the damage and fracture behavior inside rocks [16]. Studying the characteristics of rock damage evolution through acoustic emission is of great guiding significance for a deep understanding of rock fracture mechanisms and the prevention of disasters and accidents caused by rock fracture instability [17]. In recent years, achievements in using acoustic emission as an auxiliary means to study rock damage and fracture behavior have continuously emerged. This experiment will combine acoustic emission to conduct research on the damage evolution of fissured rock masses [18–23].

In order to realize the reality of the water inrush disaster process in underground and tunnel engineering and make the test results more real and reliable, the team developed a set of experimental systems that can continuously provide stable hydraulic pressure in the process of crack propagation. By taking advantage of the characteristics of high compressibility and strong pressure-retaining performance of gas, Gas, and Liquid in one chamber is used, and the liquid is driven by pressure to apply hydraulic pressure. When the water inrush channel deforms and expands (crack initiation and expansion), the water injection channel expands accordingly, and the gas-liquid common cavity in the stable hydraulic pressure device is immediately offset by gas expansion, and the gas has high compressibility, and the tiny expansion can only produce a slight pressure drop, so the permeable hydraulic pressure is relatively stable during crack initiation and expansion. Based on the fracture failure test of fissured rock mass

under stable hydraulic-stress coupling and no hydraulic action, combined with acoustic emission data collection, the fracture failure mechanical characteristics, crack propagation mechanism, and acoustic emission response mechanism of fissured rock mass in the process of water inrush in underground and tunnel engineering are systematically studied.

## 2 Stable hydraulic pressure supply test system

To realize the stable water pressure supply during the flood surge test, the stable hydraulic pressure test system was developed independently. It is mainly composed of a stable hydraulic pressure provider, flowmeter, pressure gauge, acoustic emission instrument, loading device, and control system. The whole test system is shown in Fig 1.

In the previous servo-controlled hydraulic loading system, the crack initiation and propagation were completed instantaneously during the loading process of the specimen, and the volume was expanded instantaneously, so the hydraulic pressure will drop steeply and it is difficult to achieve hydraulic stability during the crack initiation and propagation process of the rock mass. When the crack propagation is accelerated in the middle and late loading period, the volume expansion is also accelerated, and the servo response is difficult to keep up, and the servo drive has high requirements on the device response. When the servo control system receives the signal and increases the hydraulic pressure, then the crack continues to expand, causing the hydraulic pressure to drop, and the pulse-type circulating hydraulic pressure is formed repeatedly in the process of rock mass fracture, which cannot restore the actual hydraulic pressure action in the process of water inrush disaster in the tunnel and underground engineering.

## 3 Test preparation

The experiment used cement mortar material to prepare rock specimens, with a material ratio of fine sand: white cement (325 grade): water = 2:2:1 (mass ratio). The fine sand was sieved using a 1.25mm aperture sieve, and the part with an aperture less than 1.25mm was washed and dried with water to eliminate the influence of soil in the sand on the experiment. The mold used for specimen preparation is assembled with stainless steel and acrylic plates, with an internal size of length × width × height = 150mm × 50mm × 200mm. When preparing specimens with water injection holes penetrating cracks, 3D printing technology was used to prefabricate stainless steel water injection rods with different inclination angles at the ends. The ends of the rods were bonded with glue and steel sheets. After waiting for the initial setting of the specimen, first pull out the water injection rod, and then pull out the steel sheet to form a through fissure with water injection holes. The schematic diagram of the mold and the 3D specimen are shown in Fig 2.

The steady hydraulic pressure test was carried out in strict accordance with the following steps:

1. Check the test system, inject 1/3 water into the chamber of the hydraulic pressure supply device, and open the air pump to pressurize the chamber, the air pump is an intelligent numerical control, sets the pressure value range is 1.02MPa~0.98MPa (the accuracy of the air pump is 0.2bar, that is, 0.02MPa, when the air pressure rises to 1.02MPa, the air pump automatically closes. When the air pressure is reduced to 0.98MPa, the air pump automatically opens the pressure), so that the hydraulic pressure of the water supply device chamber is stable at about 1MPa.

2. Take out the cured sample to be loaded, use a hand-held stone grinder to smooth the surface of the specimen, and increase the tightness of the rubber pad.

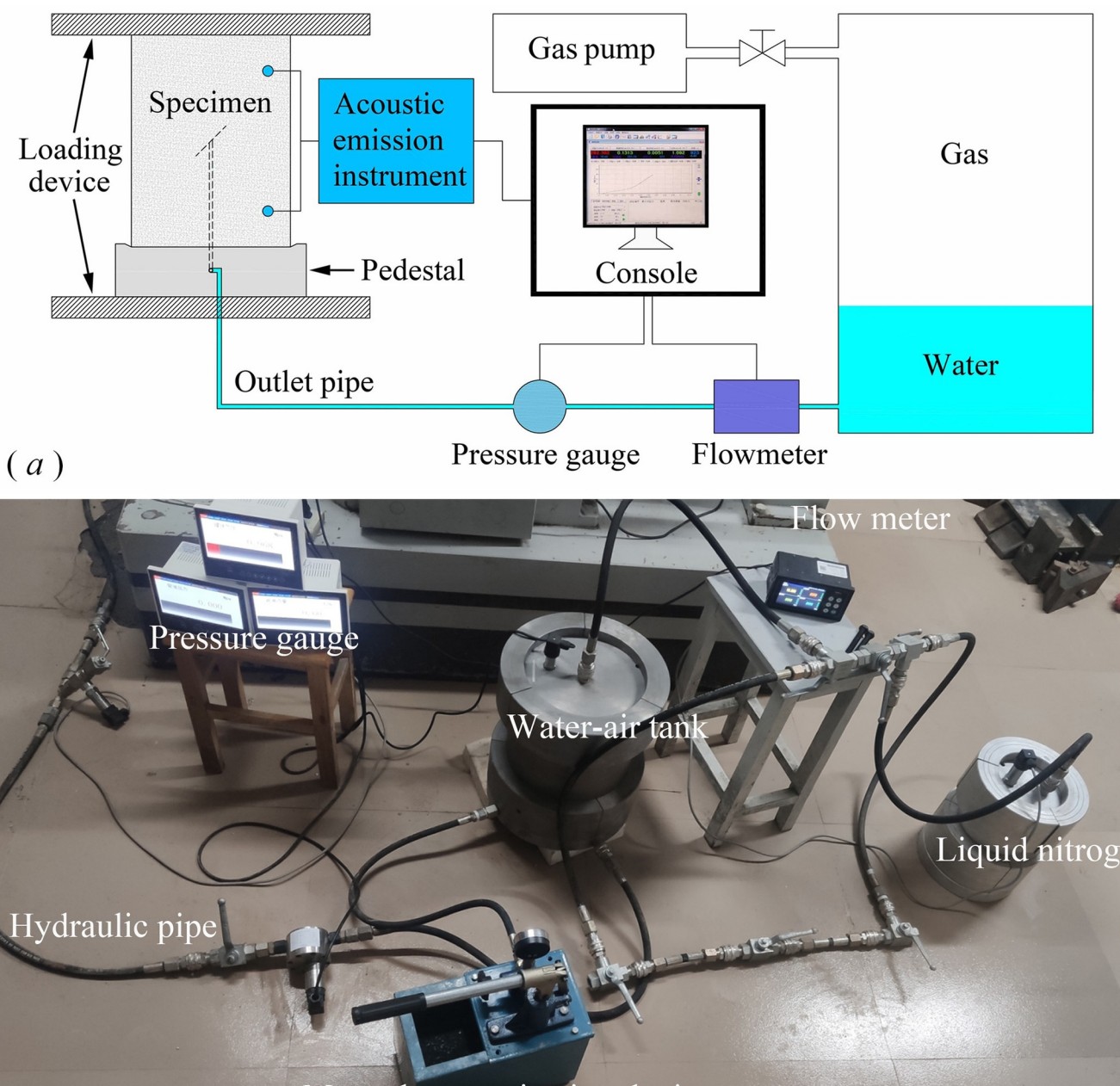

**Fig 1.** Schematic diagram of the test system (*a*) system diagram; (*b*) hydraulic pressure system.

3. Install the sealing fixture: The sealing fixture consists of two 12mm thick steel splints, two 10mm thick transparent acrylic plates, four 3mm thick rubber pads, and four sets of bolts, as shown in Fig 3. First, a steel splint is placed on the bolt, acrylic plate, and rubber pad, the specimen is placed in the middle position, and then the rubber pad, acrylic plate, and steel fixture on the other side are placed in place; this point, a relatively important link is needed: At this time, a syringe is used to inject tap water from the water injection hole of the crack specimen, and observe whether the injected water overflows from the crack surface from

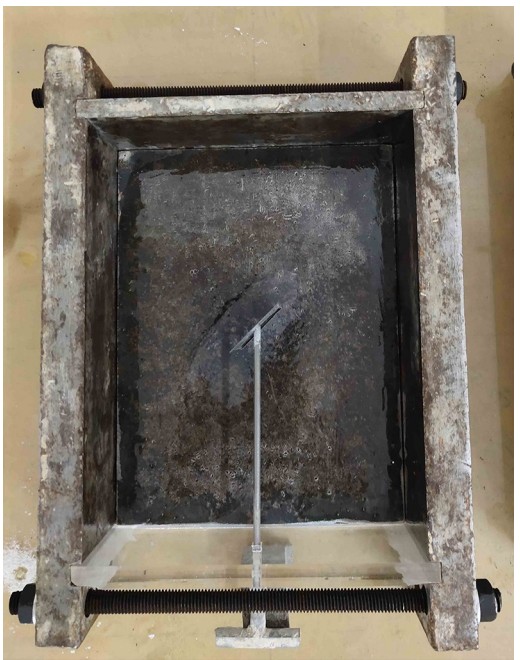

Fissure specimen

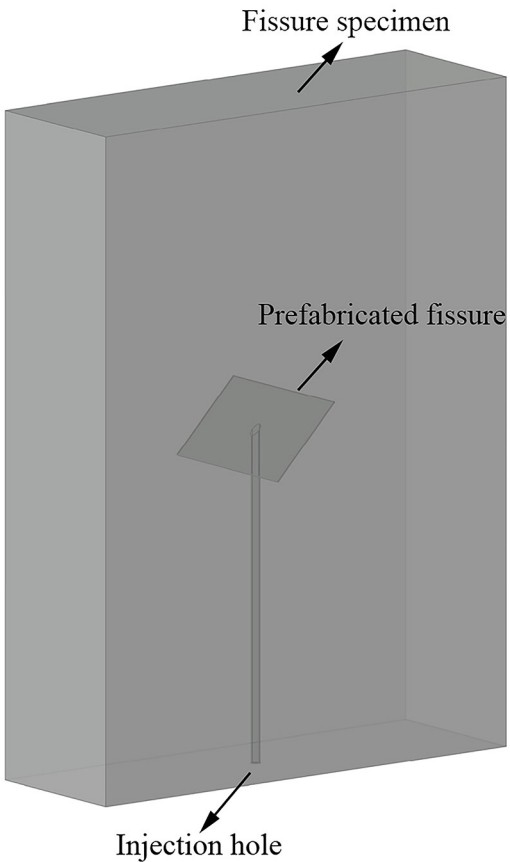

Prefabricated fissure

Injection hole

**Fig 2. Schematic diagram of the mold and the 3D specimen.**

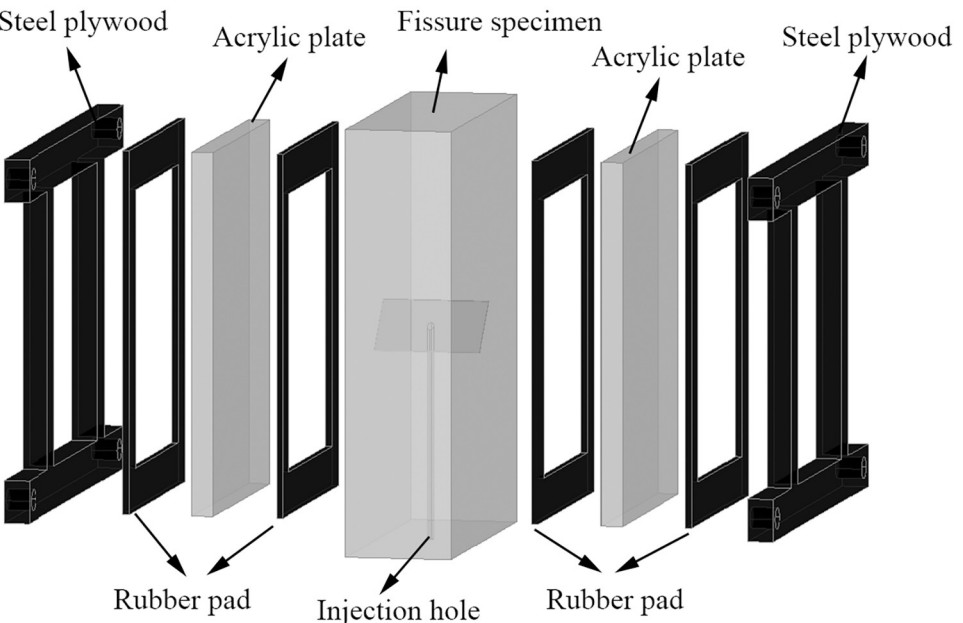

**Fig 3. CAD drawing of sealing fixture.**

the outside, so as to ensure that the water injection hole of the crack specimen is kept through with the prefabricated crack. When the hydraulic pressure is applied, the hydraulic pressure can directly act on the crack surface, and the water should fill the acrylic plate and the hollow position of the specimen as much as possible. Then the bolts were tightened with a torque wrench to keep the four bolts close to each other, and the acrylic plate was not in contact with the specimen after the installation.

4. The fixture and the specimen are placed on the loading platform as a whole, and the water injection hole is aligned with the water injection hole of the base. The sealing ring is arranged around the water injection hole of the base, and the sealing effect is achieved after the specimen is applied force; Connect the acoustic emission device to complete the acquisition software setting; Each probe was evenly coated with the coupling agent and fixed on the specimen with an acoustic emission support.

5. During the test, first apply the vertical force up to 2MPa and maintain it (loading rate is 500N/s), then turn on the hydraulic pressure switch, apply the hydraulic pressure, and then continue to apply the vertical force until the specimen is damaged. High-definition digital cameras are used throughout the test, and the failure mode is photographed after the specimen is damaged.

The loading platform of the testing machine is shown in Fig 4. You can see the support, acoustic emission probe, water sealing device, water injection quick connection, etc.

## 4 Analysis of test results

### 4.1 Hydraulic stability test results

During the hydraulic pressure stability test, 1MPa was used to stabilize the hydraulic pressure, and the uniaxial compressive stress-strain and hydraulic pressure change curves of the specimen with a built-in crack with a dip Angle of 45° were obtained, as shown in Fig 5. After the

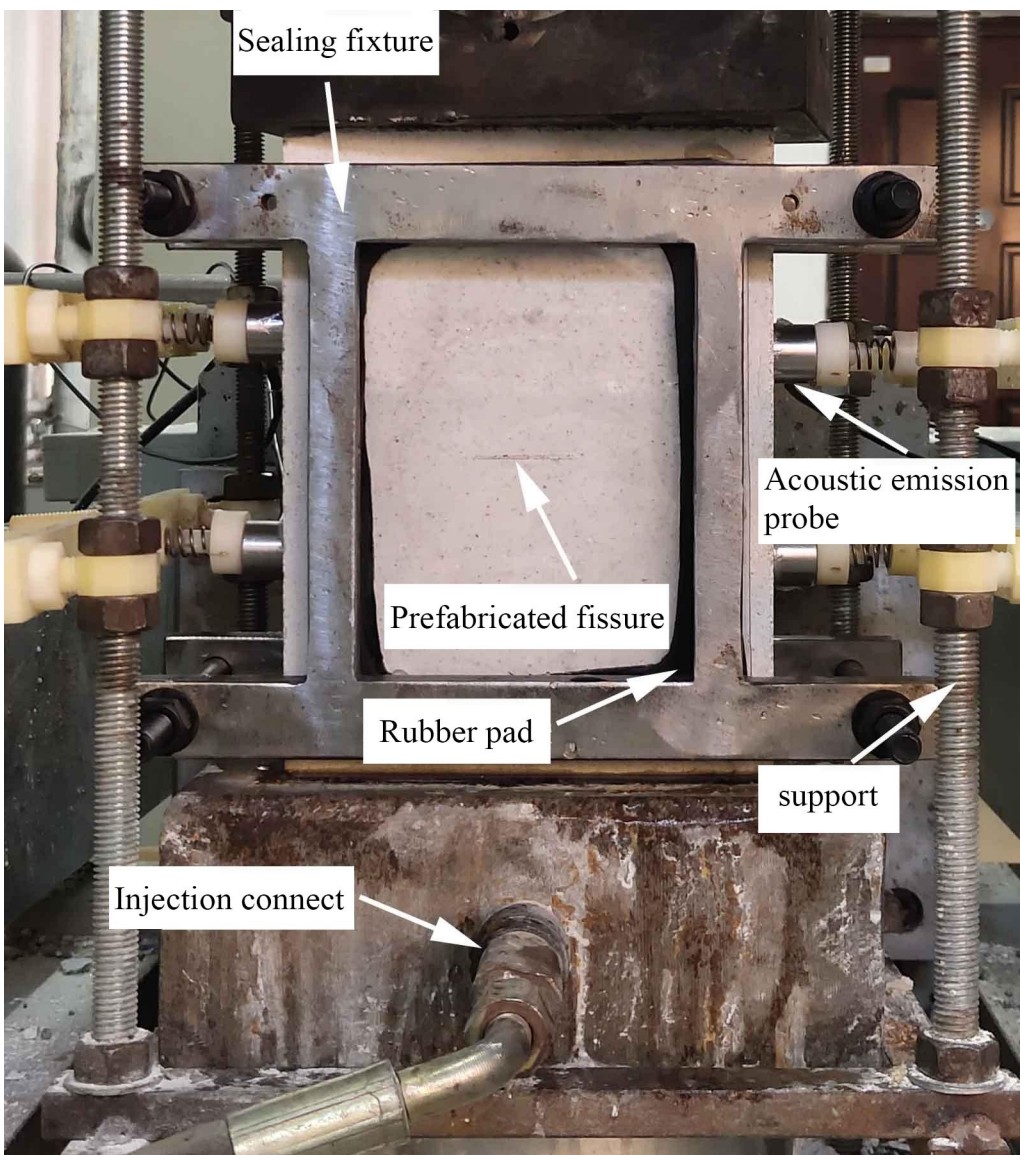

**Fig 4. Detailed drawing of testing machine loading platform.**

initial hydraulic pressure stabilized to 1001.2kPa, no hydraulic pressure was applied at the OA stage. When the applied vertical force reached 2MPa, hydraulic pressure was applied to the inside of the specimen. AB stage is the initial stage of applying stable hydraulic pressure. It can be observed from the curve of hydraulic pressure change that after applying hydraulic pressure, water enters the precast fissure through the water injection hole, and the water quantity in the hydraulic pressure chamber changes slightly, leading to pressure fluctuation. BC stage is the middle stage of applying stable hydraulic pressure. In this stage, a large number of micro-cracks are generated and expand steadily in the internal crack of the specimen, and the water steadily penetrates the new crack surface in real time and provides hydraulic pressure. At this time, the water volume in the hydraulic pressure chamber continues to decrease slightly, the gas volume continues to increase slightly, and the pressure continues to fluctuate, but the maximum fluctuation in the whole process is only 0.6kPa. About 0.06% of the hydraulic pressure

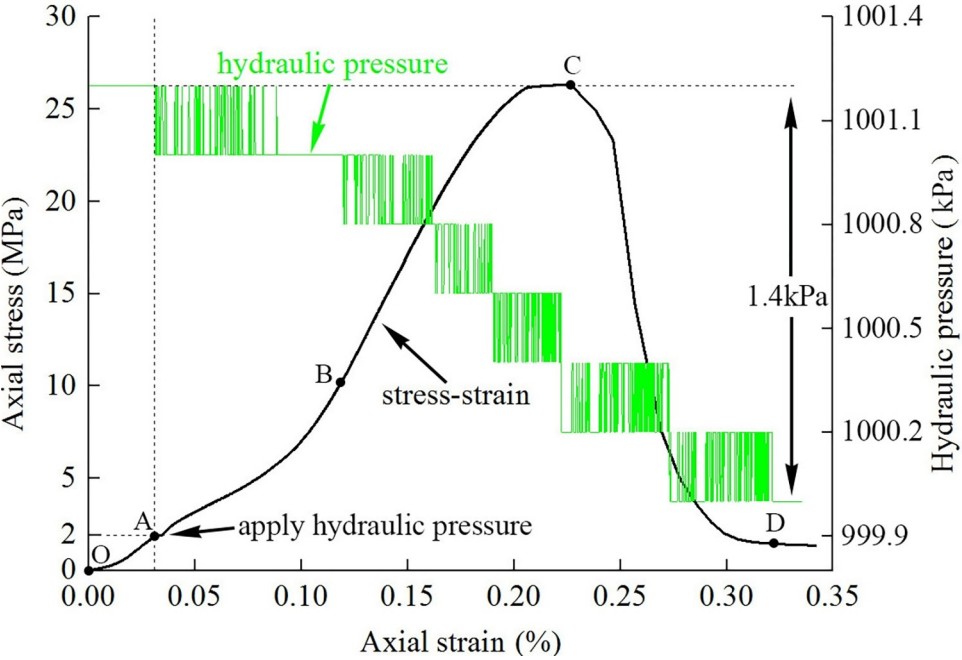

**Fig 5. Stress-strain and hydraulic pressure curves.**

of 1001kPa at that time; At stage CD, a large number of microcracks were connected in the specimen to form macroscopic cracks. At this stage, the specimen deformation was large, but the duration was short and the stress drop rate was fast. At this stage, the hydraulic pressure decreased twice. During the whole test, the hydraulic pressure decreased by 1.4kPa, which was 0.14% lower than the initial hydraulic pressure value. The hydraulic pressure fluctuated slightly and was approximately stable during the whole loading process.

## 4.2 Mechanical characteristics

To ensure that there is no water leakage between the loading base and the bottom of the specimen when the hydraulic pressure is applied, the vertical force of the crack specimen is first loaded to 2MPa, that is, the loading is suspended when the hydraulic pressure is 15KN. At this time, the hydraulic pressure device is started, the stable seepage hydraulic pressure of about 1MPa is applied, and then the vertical force is continued until the hydraulic pressure device is closed immediately after the failure of the specimen. Fig 6 shows the stress-strain curve of the crack specimen under 1MPa stable osmotic hydraulic pressure. In Fig 6, the loading curve presents obvious layering.

The peak uniaxial compressive strength of fissured rock mass with different dip angles under the action of no hydraulic pressure and 1MPa stable osmotic pressure is plotted in Fig 7 (The peak intensity is taken as the average of six sets of data). The graphic results show that the peak uniaxial compressive strength of the crack specimen increases with the increase of crack dip Angle when 1MPa is applied to stabilize the seepage pressure, and the peak strength is the lowest when the crack Angle is 0˚. By comparing the peak strength characteristics of fissured rock mass under uniaxial compression without hydraulic pressure and under 1MPa stable osmotic pressure, it is found that the peak strength of fissured rock under stable osmotic pressure is generally higher than that of specimens without hydraulic pressure under the same dip Angle, and the smaller the dip Angle, the larger the gap. The variation law of peak strength

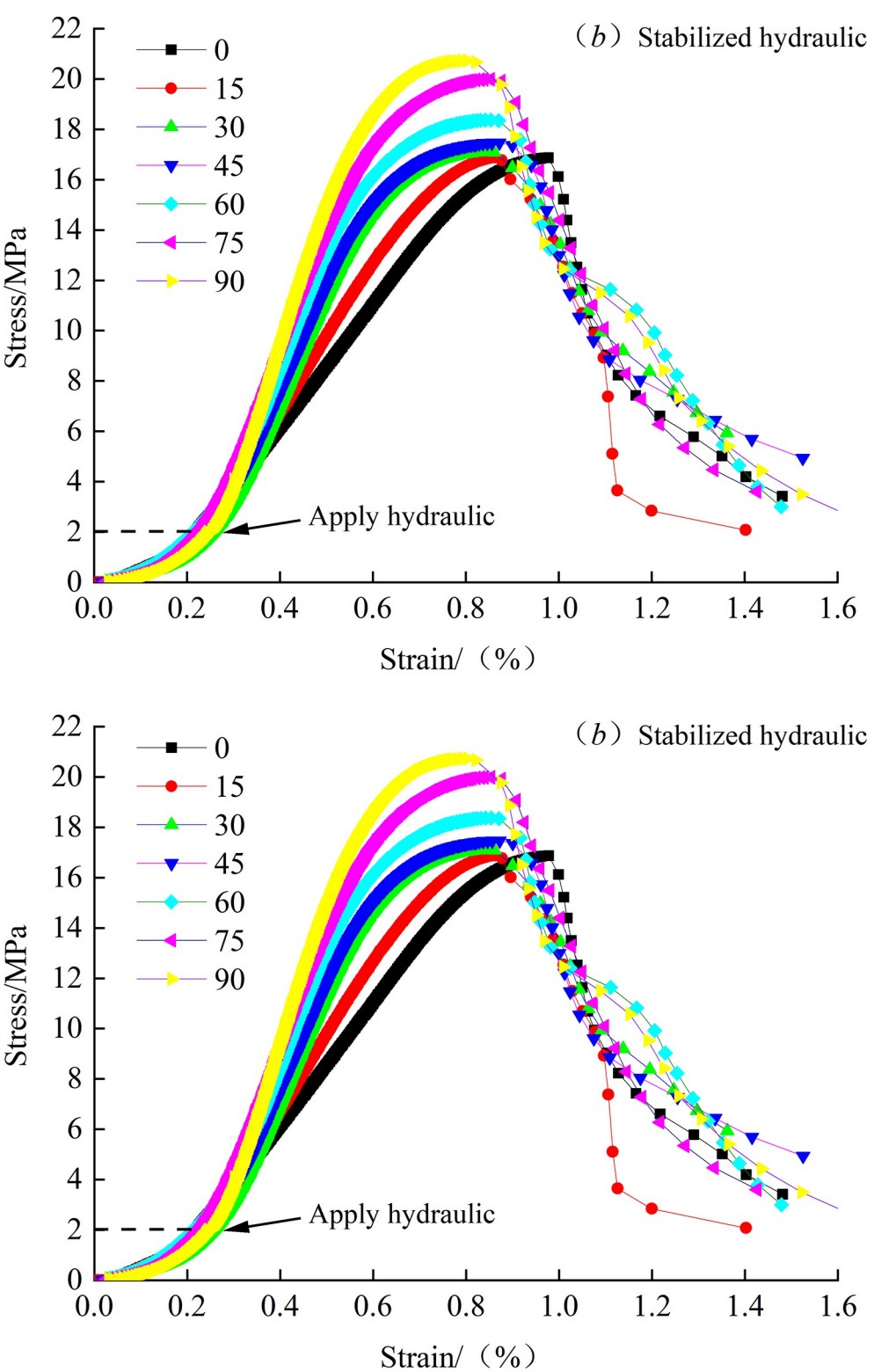

**Fig 6. Stress-strain curves with non-hydraulic and stabilized hydraulic.**

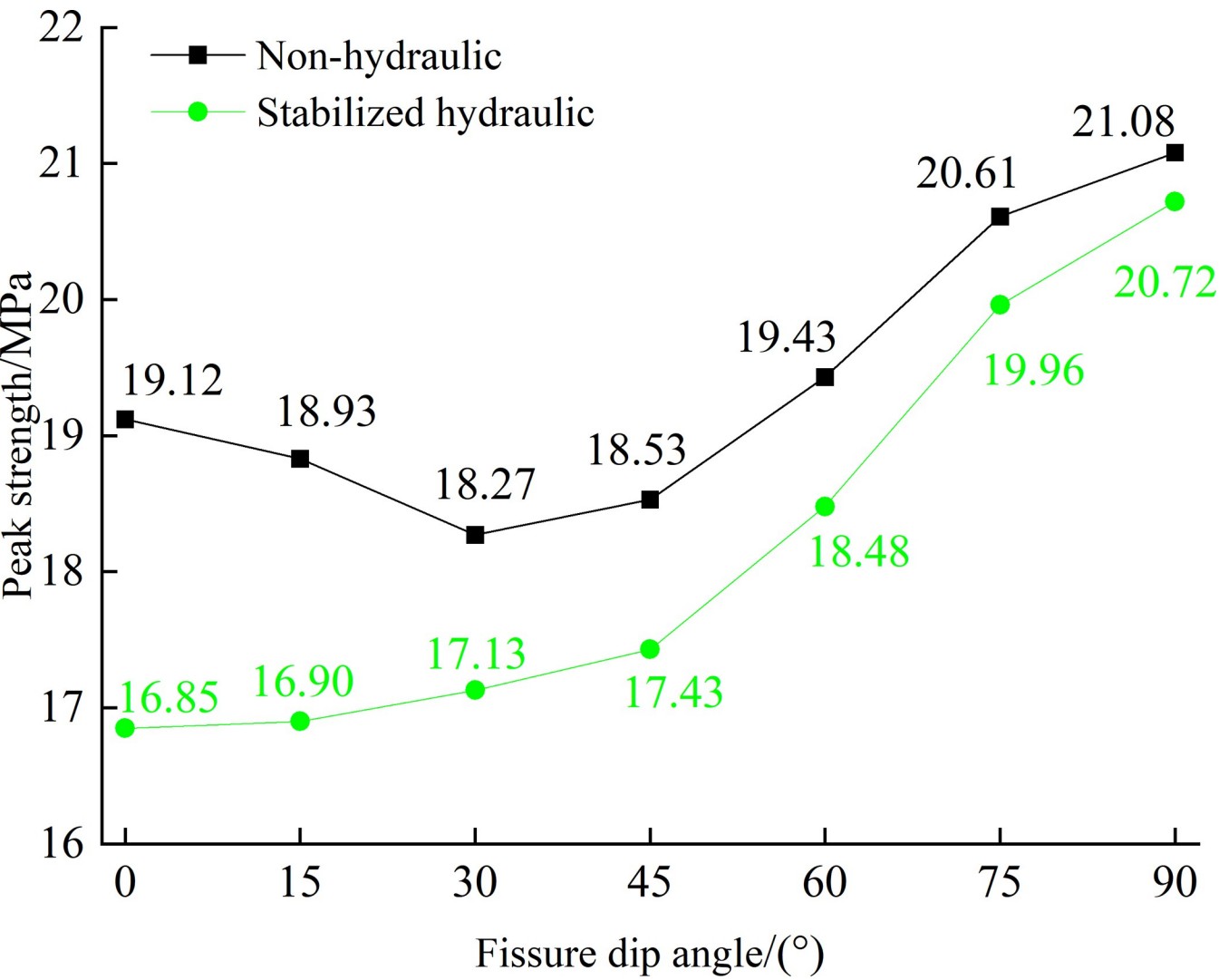

**Fig 7. Curve of peak strength change of specimen with non-hydraulic and stabilized hydraulic.**

with the increase of crack Angle also changes from first decreasing and then increasing to increasing. On the one hand, after the injection of stable hydraulic pressure into the precast crack of the fissured rock mass in the test, the precast crack surface and a part of the surface of the specimen (within the range of the rubber pad of the liquid sealing device) have a softening effect on the fissured rock mass and reduce the bearing capacity of the fissured rock mass. On the other hand, the hydraulic pressure generates continuous tensile stress on the prefabricated crack surface, which promotes the initiation of tension cracks near the prefabricated crack surface. After the crack initiation, the hydraulic pressure device provides continuous and stable hydraulic pressure and synchronously enters the new crack surface at the same time of the crack initiation, which increases the driving force of tensile crack propagation. The peak compressive strength of fissured rock mass decreases as crack propagation and interpenetration are accelerated. Finally, the change law of the peak strength of fissured rock mass with the increase of crack dip Angle under the action of hydraulic pressure is different from that without the action of hydraulic pressure. In this case, the smaller the crack dip Angle is, the greater the influence of hydraulic pressure on the peak strength of fissured rock mass.

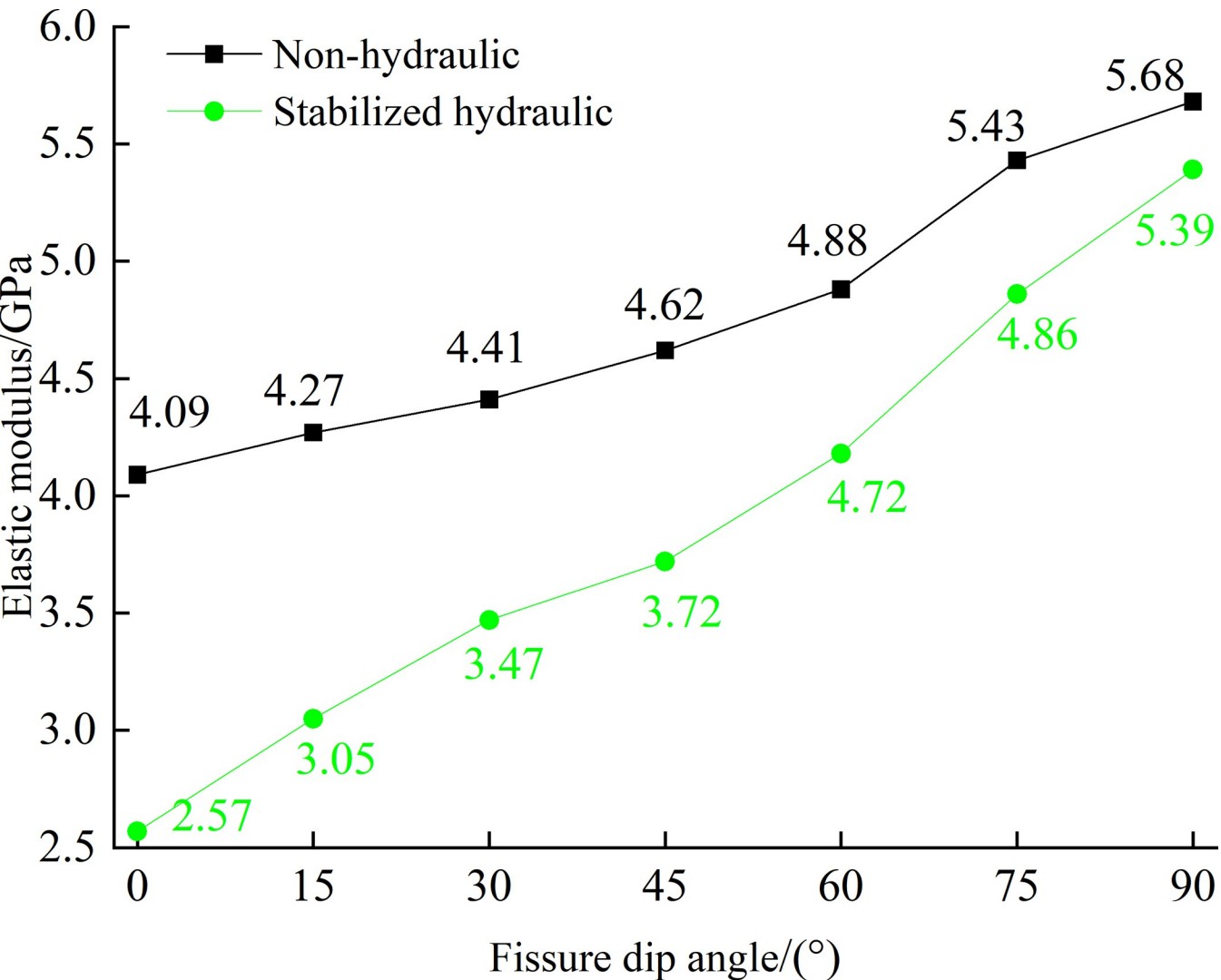

**Fig 8. Elastic modulus change curve with non-hydraulic and stabilized hydraulic.**

The average elastic modulus of crack specimens at each dip Angle under the action of no hydraulic pressure and 1MPa hydraulic pressure was plotted in Fig 8. Under the action of hydraulic pressure, the peak strength of fissured rock mass increases with the increase of crack dip Angle, which is the same as that without hydraulic pressure. However, the larger the crack Angle is, the smaller the curve spacing is. The smaller the dip Angle, the larger the spacing. The strain of fissured rock mass under the same load is larger and the elastic modulus is smaller than that without hydraulic pressure.

### 4.3 Initial crack growth state

The effect of hydraulic pressure not only affects the macroscopic mechanical characteristics such as peak strength and elastic modulus of fissured rock mass but also affects the stable hydraulic pressure when crack initiation and propagation of fissured rock mass. In the experiment, high-definition digital cameras were used for video acquisition, and the initial crack was intercepted when it expanded to the maximum path. Meanwhile, CAD was used to draw its

**Table 1. Initial crack propagation table of fissured rock mass (unit: mm).**

| Non-hydraulic | | Stabilized hydraulic | |
|---|---|---|---|
| Test result | Diagrammatic sketch | Test result | Diagrammatic sketch |
| Non-hydraulic-0˚-Test result | Non-hydraulic-0˚-Diagrammatic sketch | Stabilized hydraulic-0˚-Test result | Stabilized hydraulic-0˚-Diagrammatic sketch |
| Non-hydraulic-15˚-Test result | Non-hydraulic-15˚-Diagrammatic sketch | Stabilized hydraulic-15˚-Test result | Stabilized hydraulic-15˚-Diagrammatic sketch |
| Non-hydraulic-30˚-Test result | Non-hydraulic-30˚-Diagrammatic sketch | Stabilized hydraulic-30˚-Test result | Stabilized hydraulic-30˚-Diagrammatic sketch |
| Non-hydraulic-45˚-Test result | Non-hydraulic-45˚-Diagrammatic sketch | Stabilized hydraulic-45˚-Test result | Stabilized hydraulic-45˚-Diagrammatic sketch |
| Non-hydraulic-60˚-Test result | Non-hydraulic-60˚-Diagrammatic sketch | Stabilized hydraulic-60˚-Test result | Stabilized hydraulic-60˚-Diagrammatic sketch |
| Non-hydraulic-75˚-Test result | Non-hydraulic-75˚-Diagrammatic sketch | Stabilized hydraulic-75˚-Test result | Stabilized hydraulic-75˚-Diagrammatic sketch |
| Non-hydraulic-90˚-Test result | Non-hydraulic-90˚-Diagrammatic sketch | Stabilized hydraulic-90˚-Test result | Stabilized hydraulic-90˚-Diagrammatic sketch |

contour and calculate the crack growth path length proportionally (the length between the crack initiation point and the crack growth endpoint). The drawn uniaxial compression crack initiation mode and crack growth path length of the fissured rock mass under the action of no hydraulic pressure and 1MPa stable osmotic pressure are shown in Table 1. In the process of uniaxial compression of fissured rock mass, the initial initiation crack is generally tensile, and with the increase of vertical stress, the initial crack will continue to propagate but will stop propagate after a certain extent due to insufficient tensile stress at the initial crack tip.

As shown in Table 1, the initial crack propagation mode and path length of the fissured rock mass show that the initial crack propagation length of the fissured rock mass under stable osmotic pressure is significantly greater than that of the specimen without hydraulic pressure. During the test, the stable hydraulic pressure acted steadily on the precast fissure surface, and when the initial crack of the fissured rock mass was initiated, the stable hydraulic pressure immediately poured into the new crack surface. On the one hand, the initial crack was continuously subjected to the tensile stress of the hydraulic pressure, and the initial crack growth was mainly affected by the tensile stress. Therefore, the stable hydraulic pressure promoted the development and growth of the initial crack. On the other hand, the softening effect of the hydraulic pressure on the crack specimen reduced the tensile fracture ability of the fissure rock mass and the threshold of the initial crack initiation. Compared with the initial crack development of the fissured rock mass under different dip angles, the crack propagation length of low-dip fissured rock mass is generally higher than that of the crack specimens with large dip angles, and the propagation path also results in a larger area of hydraulic pressure and a more obvious effect. In terms of mechanical characteristics, the gap between peak strength and elastic modulus is larger at low dip angles.

## 4.4 Acoustic emission characteristics

**4.4.1 Acoustic emission ringing count.** According to the acoustic emission test results and combined with the time stress data of the test, the variation curves of acoustic emission ringing count and total ringing count (total ringing count) in the loading process of the fissure rock mass were drawn, as shown in Fig 9 (specimens with 0˚, 45˚, and 90˚ crack inclination angles were mainly taken for analysis).

As shown in Fig 9, the acoustic emission ringing count of fissured rock mass under different times and stress can be roughly divided into four stages: The first stage is in the initial compaction stage of loading, and the stress is roughly 0~30% of the peak strength. In this stage, the micro-pores inside the fissured rock mass are compressed, while the fissured rock mass made of cement mortar contains more micro-pores. With the closure of the micro-pores, weak elastic waves are released continuously, and more acoustic emission ringing counts of medium

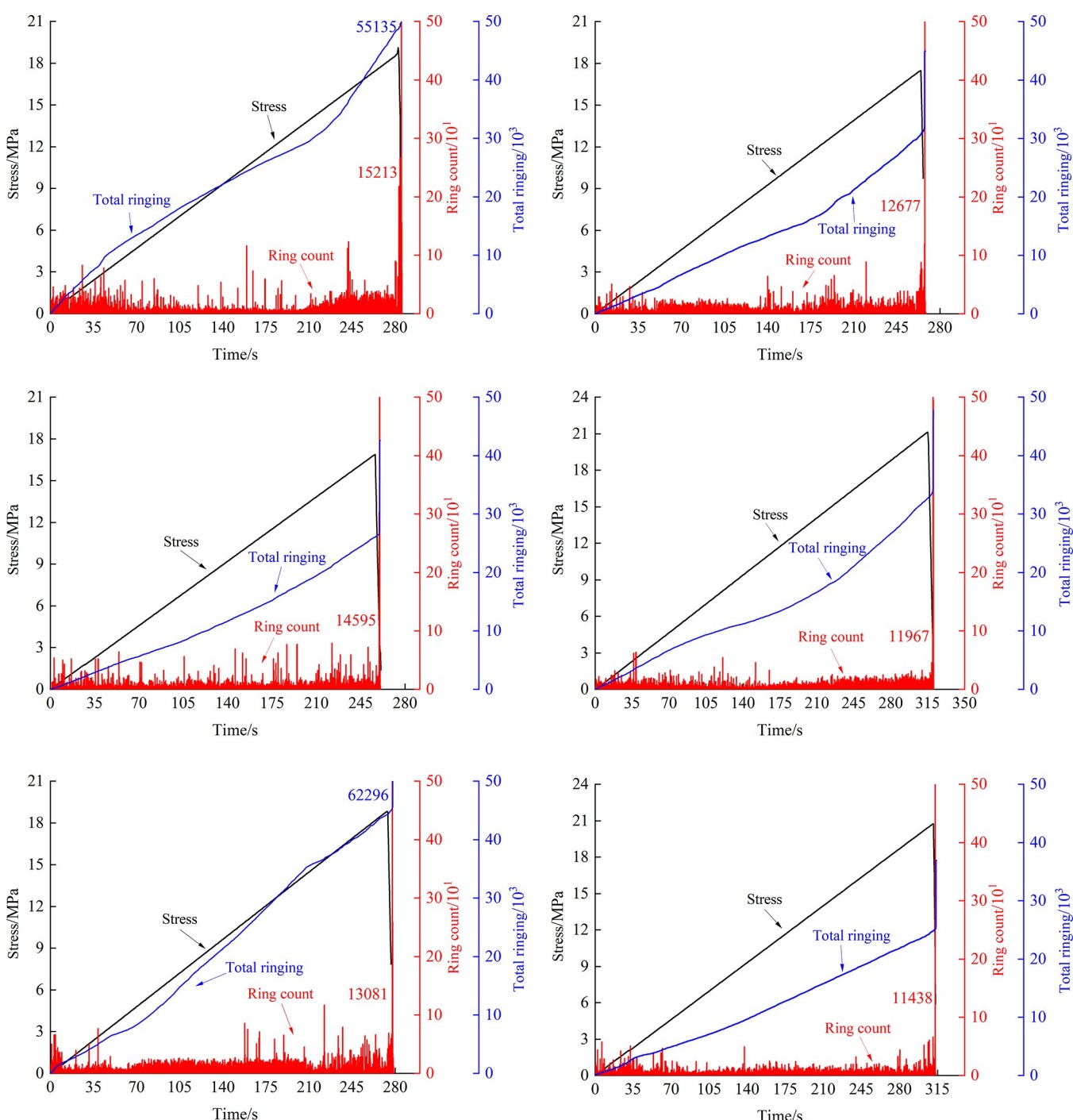

**Fig 9. Characteristic curve of acoustic emission ringing count and its total number change.** (*a*) Non-hydraulic-0˚. (*b*) Stabilized hydraulic-0˚. (*c*) Non-hydraulic-45˚. (*d*) Stabilized hydraulic-45˚. (*e*) Non-hydraulic-90˚. (*f*) Stabilized hydraulic-90˚.

and low frequency appear. The second stage is the elastic deformation stage, in which the stress is roughly 30%~50% of the peak strength, most of the micropores inside the fissured rock mass are compressed, the acoustic emission ringing count is stable without large fluctuation, and there is almost no obvious macroscopic fracture behavior inside the rock mass. The third

stage is the plastic deformation stage of crack initiation and propagation. At this stage, crack initiation and propagation begin to occur in the fissure rock mass, and there are more middle and high-frequency acoustic emission ringing counts. The fourth stage is the fissure rock mass failure stage, which is near the peak stress, cracks in the fissure rock mass accelerate expansion, mutual penetration leads to final failure, crack surfaces friction with each other, and acoustic emission ringing count rate suddenly increases sharply at the failure moment.

From the cumulative total number of acoustic emission ringing counts, it can be seen that during the loading process, the total number of acoustic emission ringing in the fissure rock mass subjected to stable hydraulic pressure is significantly lower than that in the specimen without hydraulic pressure, and the trend is also relatively stable. During the uniaxial loading process of the fissure rock mass, the acoustic emission ringing count mainly comes from the internal microporous gap compaction closure in the early stage of loading, the crack initiation and propagation in the middle and later stages, and the mutual friction between cracks. The hydraulic pressure acts on the prefabricated crack surface, and as the crack initiation and propagation enter the new crack surface, the acoustic emission ringing count generated by the mutual dislocation friction between cracks is reduced, weakening the acoustic emission signal when the fissure rock mass breaks.

**4.4.2 Peak acoustic emission frequency.** In the loading process, the peak acoustic emission frequency is the maximum amplitude value of the acoustic emission signal waveform, which is directly related to the size of acoustic emission events. The AE data results of the fissure rock mass with angles of 0˚, 45˚, and 90˚ were taken to draw the variation curve of peak frequency in the loading process of the fissure rock mass, as shown in Fig 10. The acoustic emission signals generated by tension rupture have the characteristics of short waveform, short rise time, and high frequency, while the acoustic emission signals generated by shear rupture have the characteristics of long waveform, long rise time, and low frequency [24]. He Manchao et al. [25] believe that the complexity of frequency components indicates the occurrence of multiple rupture modes.

The peak frequency of the fissured rock mass with no hydraulic pressure and stable osmotic pressure is mainly concentrated in the range of 0~100kHz, followed by the range of 100 ~200kHz, and only a few frequency signals appear in the region above 200kHz. Compared with the fissured rock mass under hydraulic pressure, there are more peak frequency signals in the range of 100kHz~200kHz in the early loading stage without hydraulic pressure, and the distribution of acoustic emission peak frequency signals is more uniform in the loading process of the fissured rock mass under stable hydraulic pressure. These phenomena indicate that the fracture morphology of fractured rock masses is more complex under no water pressure than under stable water pressure.

## 5 Discussion

### 5.1 Damage model

Acoustic emission ringing count is one of the characteristic parameters that can better reflect changes in material properties, as it is proportional to the strain energy released by the movement, fracture, and crack propagation of dislocations in the material [26]. This article describes the damage evolution characteristics of fissured rock masses using acoustic emission ringing count and cumulative ringing count as characteristic parameters.

The damage variable can be defined as

$$D = \frac{A_d}{A} \qquad (1)$$

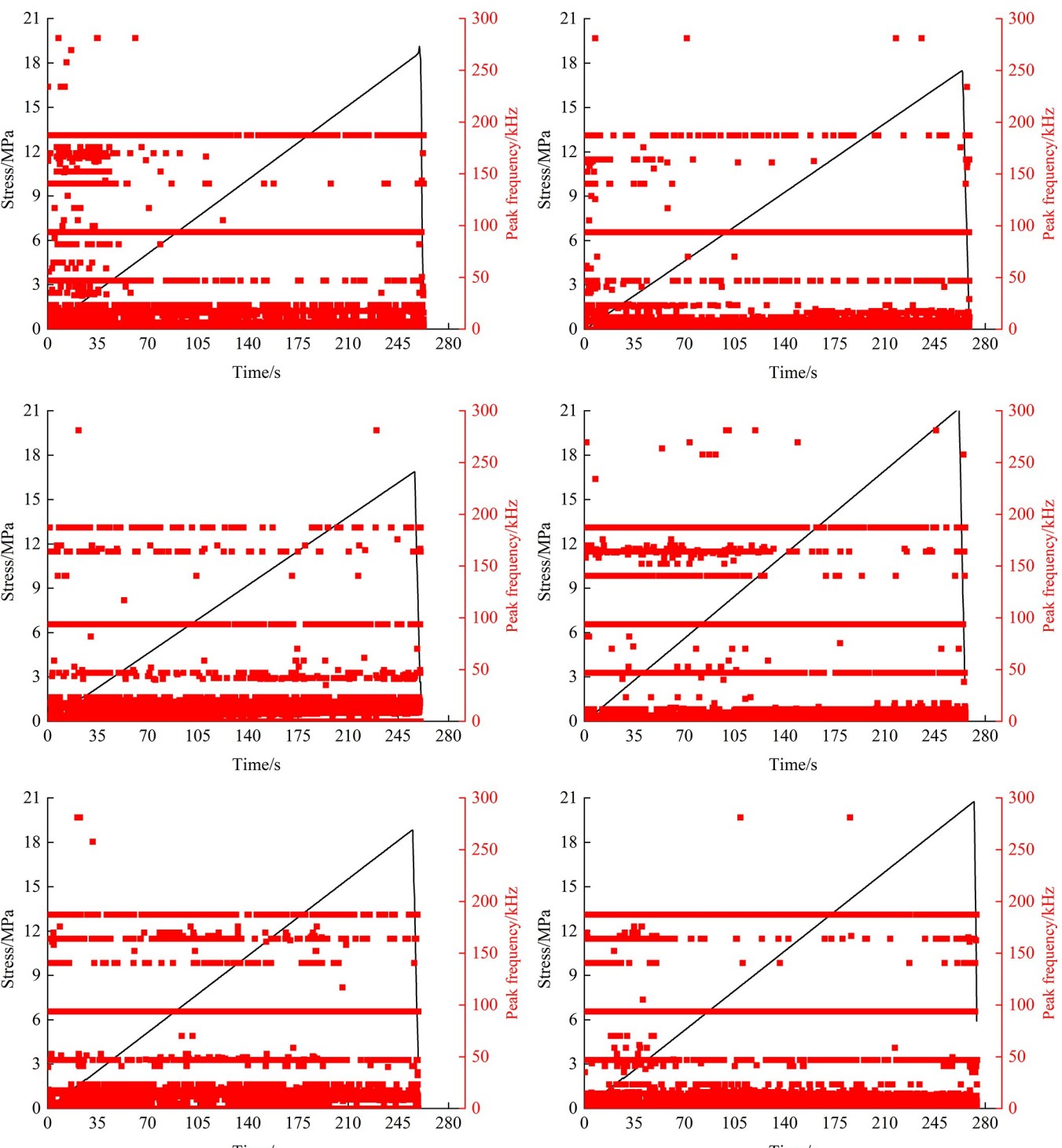

**Fig 10. Acoustic emission peak frequency variation curve.** (*a*) Non-hydraulic-0˚. (*b*) Stabilized hydraulic-0˚. (*c*) Non-hydraulic-45˚. (*d*) Stabilized hydraulic-45˚. (*e*) Non-hydraulic-90˚. (*f*) Stabilized hydraulic-90˚.

In the formula: $A_d$ is the cross-sectional area of the rock sample that undergoes compaction, the generation, propagation, convergence, and penetration of new cracks, until macroscopic damage occurs; $A$ is the initial cross-sectional area without damage. If the cumulative acoustic emission ringing count for complete failure of the entire cross-section $A$ of the non-destructive material is $C_0$, then the acoustic emission ringing count $C_w$ for micro element failure per unit area is

$$C_w = \frac{C_0}{A} \qquad (2)$$

When the cross-sectional damage area reaches $A_d$, the cumulative acoustic emission ringing count $C_d$ is

$$C_d = C_w = \frac{C_0}{A} A_d \qquad (3)$$

Therefore, there are

$$D = \frac{C_d}{C_0} \qquad (4)$$

## 5.2 Damage evolution analysis

According to the damage variable defined by the acoustic emission ringing count in Eq (4), combined with the acoustic emission ringing count in the experiment, the curve of the damage variable changes with loading time, as shown in Fig 11. From the graphical results, it can be seen that the damage evolution process of fissured rock mass during loading can be roughly divided into four stages, as shown in Fig 12.

Stage 1: This stage is mainly the stage of pore compaction inside the fissured rock mass. Due to the fact that the rock mass material prepared with cement mortar has more internal pores compared to the real rock, some acoustic emission events will occur during this stage, leading to damage.

Stage 2: This stage is the elastic damage stage, and the fissured rock mass is mainly in the elastic deformation stage. The increase in internal damage is approximately linear, and the acoustic emission events are relatively stable.

Stage 3: During this stage, macroscopic cracks gradually emerge in the fissured rock mass, and the damage to the rock mass gradually reaches a critical value. The cracks continue to initiate and expand, and the acoustic emission activity gradually becomes active. However, compared to specimens without hydraulic pressure, the fissured rock mass under stable hydraulic pressure is affected by the action of permeable water. During the process of crack initiation and expansion, permeable water can enter new crack surfaces, reducing the number of acoustic emission events and making the damage growth of the fissured rock mass similar to that of the elastic damage stage.

Stage 4: This stage is the damage and failure stage. Due to the rapid accumulation of damage to the specimen, the number of acoustic emission events increases significantly, and the damage variable suddenly increases, causing the specimen to fail.

From the evolution process of acoustic emission damage in rocks, it can be seen that the initiation and evolution of cracks and damage, stable development until the occurrence of large-scale crack propagation, and then the entire process from crack propagation to macroscopic fracture of the rock sample. Using the cumulative ringing count of acoustic emission as a characterization parameter to analyze the damage evolution and failure of rocks can better reflect the gradual evolution process from the initiation and propagation of internal cracks to failure.

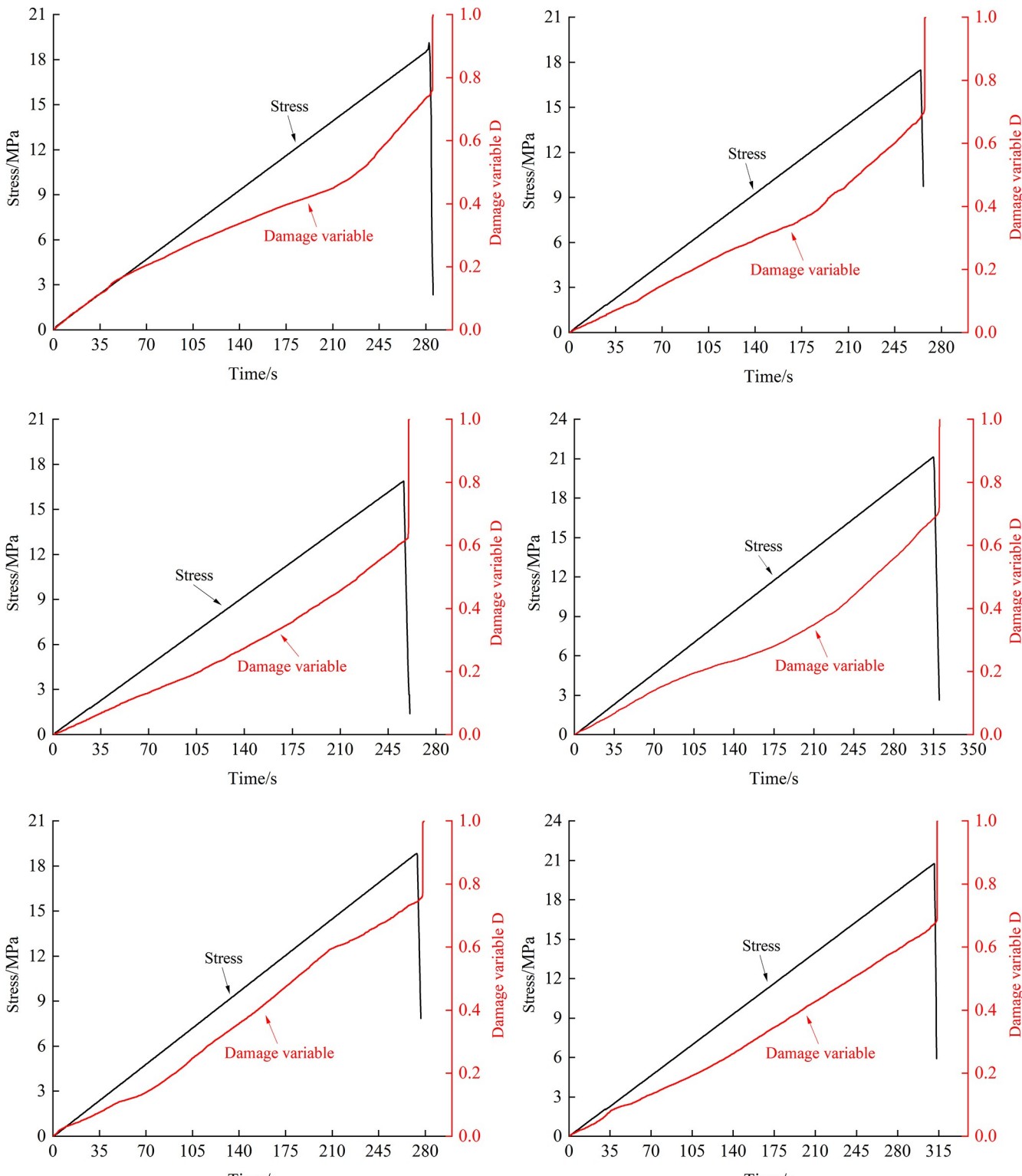

**Fig 11. Damage variable evolution curve.** (*a*) Non-hydraulic-0˚. (*b*) Stabilized hydraulic-0˚. (*c*) Non-hydraulic-45˚. (*d*) Stabilized hydraulic-45˚. (*e*) Non-hydraulic-90˚. (*f*) Stabilized hydraulic-90˚.

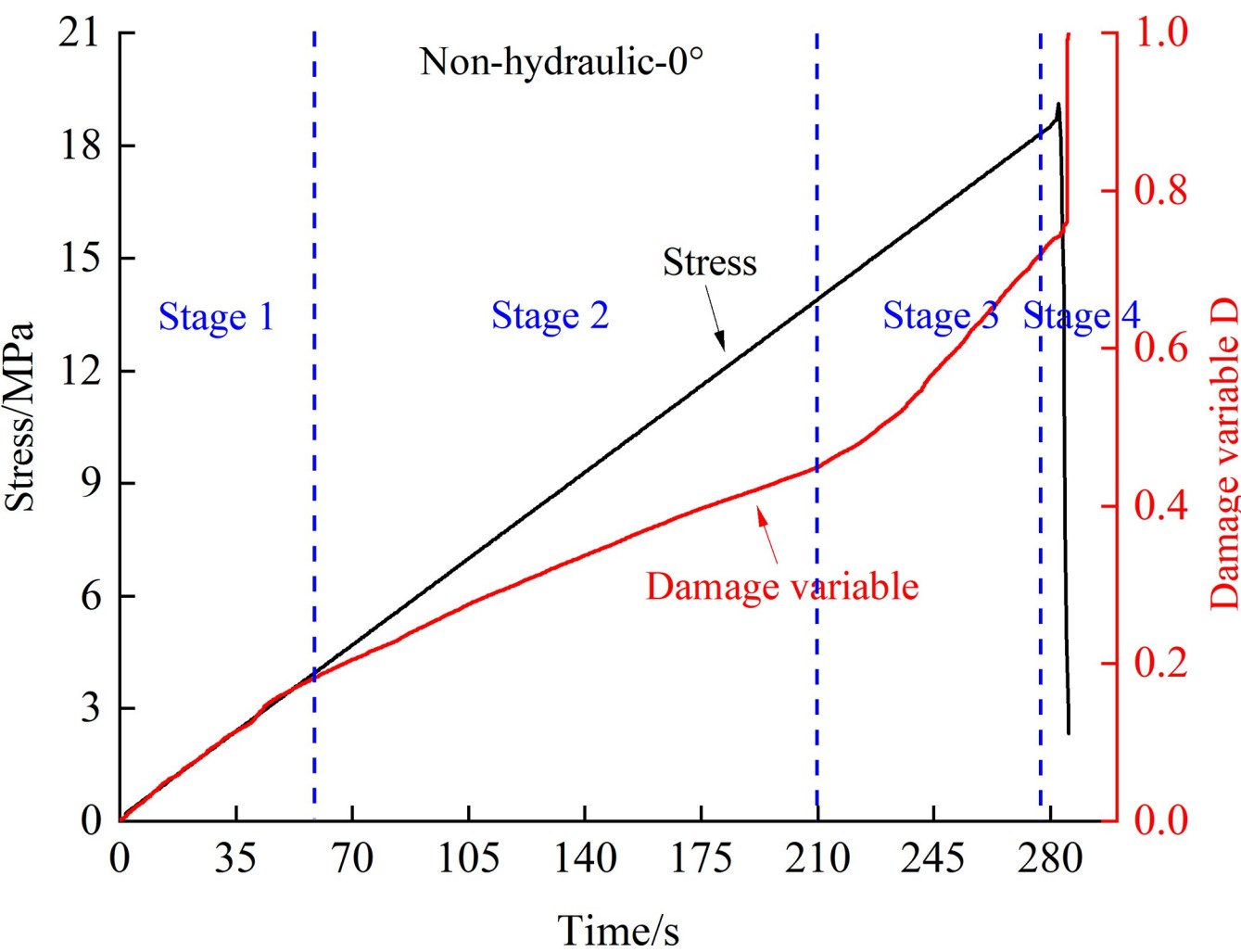

**Fig 12. Characteristic curve of damage variable stage.**

From the curve of damage variables, it can be seen that under stable hydraulic pressure, the changing trend of damage variables in the fissured rock mass is more stable. In the absence of hydraulic pressure, the variation trend of damage variables fluctuates more greatly, and the rate of damage increase is also unstable, indicating that the failure mode of fissure specimens without water pressure is more complex. This conclusion is consistent with the results exhibited by peak frequency.

## 6 Conclusion

1. Independently developed a test system that can continuously and stably provide hydraulic pressure in the process of crack initiation and propagation of fissured rock mass, and verified by the characteristic curve of hydraulic pressure variation during the experiment.

2. Under the effect of stable hydraulic pressure, the change law of peak strength of fissured rock mass with the increase of crack Angle changes from first decreasing and then increasing in the absence of hydraulic pressure to increasing law, and the peak strength and elastic modulus are lower than those of fissured rock mass without hydraulic pressure.

3. The crack propagation length of low-dip fissured rock mass is generally higher than that of large-dip fissured rock specimens. The propagation path length increases the range and effect of hydraulic pressure, and the initial crack propagation length of the fissured rock mass under hydraulic pressure is also significantly greater than that of the specimens without hydraulic pressure.

4. The acoustic emission ringing count and damage variable can be roughly divided into four stages. From the cumulative total number of acoustic emission ringing counts, the total number of acoustic emission ringing in fissured rock masses subjected to hydraulic pressure is significantly lower than that in specimens without hydraulic pressure, and the trend is also relatively stable.

## Supporting information

**S1 Dataset.**
(DOCX)

**S2 Dataset.**
(DOCX)

**S1 Table.**
(ZIP)

## Author Contributions

**Conceptualization:** Jiajun Zeng, Chengzhi Pu.

**Data curation:** Qingqing Shen.

**Formal analysis:** Jiajun Zeng, Qingqing Shen.

**Funding acquisition:** Qiyun Wang.

**Investigation:** Qiang Zeng, Zhicheng Yang.

**Methodology:** Jiajun Zeng, Chengzhi Pu.

**Validation:** Qiyun Wang.

**Writing – original draft:** Jiajun Zeng.

**Writing – review & editing:** Jiajun Zeng, Chengzhi Pu.

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
