## [Decision Letter · Decision Letter 0]

20 May 2024

PONE-D-24-14746Tunnel and underground engineering rock mass water inrush damage and acoustic emission characteristicsPLOS ONE

Dear Dr. Zeng,

Thank you for submitting your manuscript to PLOS ONE. After careful consideration, we feel that it has merit but does not fully meet PLOS ONE’s publication criteria as it currently stands. Therefore, we invite you to submit a revised version of the manuscript that addresses the points raised during the review process.

Please, carefully follow the reviewer's suggestions and properly answer to the raised points.

We look forward to receiving your revised manuscript.

Kind regards,

Fabio Trippetta, Ph.D.

Academic Editor

PLOS ONE

“This work is supported by Hunan Provincial Natural Science Foundation of China (2023JJ50104); Scientific Research Fund of Hunan Provincial Education Department (21B0803, 22B0853, 23B0838); Hunan Institute of Technology provincial-level applied characteristic discipline (KFB23022); National College Student Innovation and Entrepreneurship Training Program Project(S202311528042).”

“The authors appreciate the constructive comments from the anonymous reviewers. This work is supported by Hunan Provincial Natural Science Foundation of China (2023JJ50104); Scientific Research Fund of Hunan Provincial Education Department (21B0803, 22B0853, 23B0838); Hunan Institute of Technology provincial-level applied characteristic discipline (KFB23022); National College Student Innovation and Entrepreneurship Training Program Project(S202311528042).”

“This work is supported by Hunan Provincial Natural Science Foundation of China (2023JJ50104); Scientific Research Fund of Hunan Provincial Education Department (21B0803, 22B0853, 23B0838); Hunan Institute of Technology provincial-level applied characteristic discipline (KFB23022); National College Student Innovation and Entrepreneurship Training Program Project(S202311528042).”

Reviewers' comments:

Reviewer's Responses to Questions

**Comments to the Author**

1. Is the manuscript technically sound, and do the data support the conclusions?

Reviewer #1: Yes

Reviewer #2: Partly

2. Has the statistical analysis been performed appropriately and rigorously? 

Reviewer #1: Yes

Reviewer #2: Yes

3. Have the authors made all data underlying the findings in their manuscript fully available?

Reviewer #1: Yes

Reviewer #2: Yes

4. Is the manuscript presented in an intelligible fashion and written in standard English?

Reviewer #1: Yes

Reviewer #2: Yes

5. Review Comments to the Author

Reviewer #1: This paper conducted a fracture failure test of fissured rock mass under stable hydraulic pressure using a self-developed stable hydraulic pressure device, verified the feasibility of the test system, and analyzed the acoustic emission characteristics and damage evolution law of the fracture failure process. The research content of this article is novel, and the research results have certain guiding significance for the prevention and control of water inrush disasters in tunnels and geotechnical engineering. Suggest receiving after minor optimizations and improvements.

1. Figure 1 shows the CAD diagram of a stable infiltration water pressure system. Please provide a laboratory diagram of the test system

2. In Section of “2 Test preparation”, The CAD specimen diagram in Figure 1 can depict the dimensions, and there are duplicate labels in the diagram. Please check and correct them

3. In Section of “2 Test preparation”, “After the initial setting of the specimens, the water injection rods were pulled out and the steel sheets were embedded. ” This sentence describes some issues. I would like to know if "the steel sheets were embedded" should be pulled out or embedded at this time?

4. In Section of “Introduction” , "The rock-like materials in this test are made by mixing 425 white cement: fine sand: water at the mass ratio of 5:5:2." I'd like to know how to ensure the homogeneity of the sample ?

5. In the peak intensity of Figure 6, is the data taken as a single specimen or the average of multiple specimens? If it is multiple, please indicate the number of specimens.

6. Please unify the words "osmotic water pressure" and "water pressure" in the paper

7. Please unify the words "osmotic water pressure, water pressure, hydraulic pressure, etc." in the paper.

8. In Figure 8, the time axis is used as the horizontal axis for the acoustic emission ringing count and stress curve. Is it convenient to convert the time axis to strain, and have other studies used time as the horizontal axis to describe it?

9. The color of the right vertical axis in Figure 10 (b) is inconsistent. Please check all information and language throughout the text.

Reviewer #2: The author independently developed a stable permeability hydraulic pressure device, conducted hydraulic-stress coupling tests on fractured rock masses, analyzed the mechanical characteristics and crack initiation modes of fissured rock masses, and explored the acoustic emission and damage characteristics during the loading process. Some useful conclusions were obtained, which are helpful for the prevention and control of water inrush disasters in tunnels and underground engineering. There are still some minor issues in this article that need further optimization and improvement.

1. Introduction, page 3: “ the gas-liquid concavity is used, and the liquid is driven by pressure to apply water pressure. ” Could you please explain the meaning of “the gas-liquid concavity” here.

2. Figure 1 only shows a schematic diagram, without the actual equipment diagram during indoor testing. It is best to describe and explain it in conjunction with the actual diagram.

3. Test preparation, page 4: Figure 1 has duplicate naming. Please check all figures, tables, titles, and other serial numbers to correct any issues.

4. Test preparation, page 4: “The steady water pressure test was carried out in strict accordance with the following steps”. More precise and concise description of the experimental steps during the loading process.

5. The author's review of the latest related research on tunnel safety, rock crack propagation, and non-destructive testing is insufficient. The following latest published literature can provide reference for the author's research:

[1] Case study on the secondary support time and optimization of combined support for a roadway under high in-situ stress. Geomechanics and Geophysics for. Geo-Energy and Geo-Resource. 2024, 10(1): 66. https://doi.org/10.1007/s40948-024-00774-w

[2] Analytical solution of the stress field and plastic zone at the tip of a closed crack. Frontiers in Earth Science, 2024, 12: 1370672. https://doi.org/10.3389/feart.2024.1370672

[3] Experimental study of dynamic characteristics of tailings with different reconsolidation degrees after liquefaction. Frontiers in Earth Science, 2022, 10: 876401. https://doi.org/10.3389/feart.2022.876401

6. Different descriptive words such as “osmotic water pressure”, “osmotic water pressure”, “no hydraulic pressure”, and “non-hydraulic” appear in the text. Please check the relevant vocabulary and provide a unified description

7. In section 3.4.2, the author selected the peak frequency of acoustic emission for analysis. Please further explain the information characterizing the peak frequency of acoustic emission and explain the pattern of acoustic emission frequency.

8. Discussion, page 15: The font size of the first paragraph of this section does not match the font size of the previous and subsequent articles. Please verify and modify it.

9. Figure 10: The color of some image coordinate axes is inconsistent with other graphics. In the analysis, it should be added to analyze the differences in acoustic emission damage under two types of loads: Non-hydraulic and Stabilized hydraulic.

6. PLOS authors have the option to publish the peer review history of their article (what does this mean?). If published, this will include your full peer review and any attached files.

Reviewer #1: No

Reviewer #2: No

---

## [Author Response · Author response to Decision Letter 0]

31 May 2024

Dear Editors and Reviewers:

Thank you for your letter and for the reviewers’ comments concerning our manuscript entitled “Tunnel and underground engineering rock mass water inrush damage and acoustic emission characteristics” ( Manuscript #PONE-D-24-14746 ). Those comments are all valuable and very helpful for revising and improving our paper, as well as the important guiding significance to our researches. We have studied comments carefully and have made correction which we hope meet with approval. However, some reviewers did not provide direct comments or questions. Revised portion are marked in “Revised Manuscript with Track Changes”. The main corrections in the paper and the responds to the reviewer’s comments are as flowing:

Responds to the reviewer’s comments:

Reviewer #1: This paper conducted a fracture failure test of fissured rock mass under stable hydraulic pressure using a self-developed stable hydraulic pressure device, verified the feasibility of the test system, and analyzed the acoustic emission characteristics and damage evolution law of the fracture failure process. The research content of this article is novel, and the research results have certain guiding significance for the prevention and control of water inrush disasters in tunnels and geotechnical engineering. Suggest receiving after minor optimizations and improvements.

1. Figure 1 shows the CAD diagram of a stable infiltration water pressure system. Please provide a laboratory diagram of the test system

Response: According to the reviewer's comments, the author has placed the actual diagram of the laboratory testing device in Figure 1.

2. In Section of “2 Test preparation”, The CAD specimen diagram in Figure 1 can depict the dimensions, and there are duplicate labels in the diagram. Please check and correct them

Response: Thank you for the reviewer's reminder. The author has checked and corrected any duplicates in the article.

3. In Section of “2 Test preparation”, “After the initial setting of the specimens, the water injection rods were pulled out and the steel sheets were embedded. ” This sentence describes some issues. I would like to know if "the steel sheets were embedded" should be pulled out or embedded at this time?

Response: “After waiting for the initial setting of the specimen, first pull out the water injection rod, and then pull out the steel sheet to form a through fissure with water injection holes”

4. In Section of “Introduction” , "The rock-like materials in this test are made by mixing 425 white cement: fine sand: water at the mass ratio of 5:5:2." I'd like to know how to ensure the homogeneity of the sample ?

Response: The uniformity of each specimen is ensured through the following methods: during the preparation of specimens, the mix proportions of various materials are strictly controlled, and the materials are placed in the mixer in order. The same mixing time is set for each batch of specimens, and after mixing, they are poured into the mold and vibrated at the same speed and time to form the specimens. After demolding, place it in the same standard curing environment for curing, ensuring the uniformity of each specimen.

5. In the peak intensity of Figure 6, is the data taken as a single specimen or the average of multiple specimens? If it is multiple, please indicate the number of specimens.

Response: The peak intensity in Figure 6 is taken as the average of six sets of data, which will be explained in the corresponding positions in the text.

6. Please unify the words "osmotic water pressure" and "water pressure" in the paper

Response: Thank you to the reviewers for their careful review. The author has checked and unified the wording of the entire article.

7. Please unify the words "osmotic water pressure, water pressure, hydraulic pressure, etc." in the paper.

Response: The author has checked the entire article and made revisions to any inappropriate or inconsistent wording.

8. In Figure 8, the time axis is used as the horizontal axis for the acoustic emission ringing count and stress curve. Is it convenient to convert the time axis to strain, and have other studies used time as the horizontal axis to describe it?

Response: When analyzing acoustic emission data, strain or time is usually used as the horizontal axis, while acoustic emission collection is usually collected in units of time. When using displacement loading mode, the transformation relationship between strain and time is relatively direct, and in this case, strain is usually used as the abscissa for analysis. When using force loading mode, the transformation relationship between strain and time is non-linear, so time is usually used as the horizontal axis for analysis. As shown in the following literature:

Chen Q, Zhengyang S, Yu W, et al. Fractures and Acoustic Emission Features of Non-persistent Jointed Rocks Subjected to Freeze–Thaw-Compression Load: Experimental Insights[J]. Rock Mechanics and Rock Engineering, 2022(1):55.DOI:10.1007/s00603-021-02667-w.

9. The color of the right vertical axis in Figure 10 (b) is inconsistent. Please check all information and language throughout the text.

Response: Based on the comments of the reviewers, the author has made modifications to the positions indicated by the reviewers and checked and improved the image and language information throughout the entire text.

Reviewer #2: The author independently developed a stable permeability hydraulic pressure device, conducted hydraulic-stress coupling tests on fractured rock masses, analyzed the mechanical characteristics and crack initiation modes of fissured rock masses, and explored the acoustic emission and damage characteristics during the loading process. Some useful conclusions were obtained, which are helpful for the prevention and control of water inrush disasters in tunnels and underground engineering. There are still some minor issues in this article that need further optimization and improvement.

1. Introduction, page 3: “ the gas-liquid concavity is used, and the liquid is driven by pressure to apply water pressure. ” Could you please explain the meaning of “the gas-liquid concavity” here.

Response: The improper use of words here has caused difficulties in reading. "The gas liquid concentration" should be "Gas and liquid in one chamber". The author has made modifications.

2. Figure 1 only shows a schematic diagram, without the actual equipment diagram during indoor testing. It is best to describe and explain it in conjunction with the actual diagram.

Response: According to the reviewer's comments, the author has placed the actual diagram of the laboratory testing device in Figure 1.

3. Test preparation, page 4: Figure 1 has duplicate naming. Please check all figures, tables, titles, and other serial numbers to correct any issues.

Response: The author has made corrections to the information mentioned by the reviewer and checked the entire text.

4. Test preparation, page 4: “The steady water pressure test was carried out in strict accordance with the following steps”. More precise and concise description of the experimental steps during the loading process.

Response: Thank you for the reviewer's reminder. The step description here is too long. The author will make appropriate modifications to the description here to make it more precise and concise.

5. The author's review of the latest related research on tunnel safety, rock crack propagation, and non-destructive testing is insufficient. The following latest published literature can provide reference for the author's research:

[1] Case study on the secondary support time and optimization of combined support for a roadway under high in-situ stress. Geomechanics and Geophysics for. Geo-Energy and Geo-Resource. 2024, 10(1): 66. https://doi.org/10.1007/s40948-024-00774-w

[2] Analytical solution of the stress field and plastic zone at the tip of a closed crack. Frontiers in Earth Science, 2024, 12: 1370672. https://doi.org/10.3389/feart.2024. 1370672

[3] Experimental study of dynamic characteristics of tailings with different reconsolidation degrees after liquefaction. Frontiers in Earth Science, 2022, 10: 876401. https://doi.org/10.3389/feart.2022.876401

Response: Thank you for the reviewer's comments. I have reviewed the literature recommended by the reviewers. The following literature has studied the rheological problem of tunnel surrounding rock, which is somewhat related to the underground and tunnel engineering surrounding rock disasters investigated in the article literature. Therefore, it is used as a reference for citation.

“Case study on the secondary support time and optimization of combined support for a roadway under high in-situ stress. Geomechanics and Geophysics for. Geo-Energy and Geo-Resource. 2024, 10(1): 66. https://doi.org/10.1007/s40948-024- 00774-w”

6. Different descriptive words such as “osmotic water pressure”, “osmotic water pressure”, “no hydraulic pressure”, and “non-hydraulic” appear in the text. Please check the relevant vocabulary and provide a unified description

Response: The author has checked the entire article and made revisions to any inappropriate or inconsistent wording.

7. In section 3.4.2, the author selected the peak frequency of acoustic emission for analysis. Please further explain the information characterizing the peak frequency of acoustic emission and explain the pattern of acoustic emission frequency.

Response: Thank you for the reviewer's comments. The author has improved the meaning of acoustic emission frequency and the information expressed in this type of acoustic emission frequency in the article.

8. Discussion, page 15: The font size of the first paragraph of this section does not match the font size of the previous and subsequent articles. Please verify and modify it.

Response: Thank you for the careful review by the reviewer. The author has made corrections to the font and formatting errors in the text.

9. Figure 10: The color of some image coordinate axes is inconsistent with other graphics. In the analysis, it should be added to analyze the differences in acoustic emission damage under two types of loads: Non-hydraulic and Stabilized hydraulic.

Response: The author has made changes to the coordinate information based on the comments of the reviewers. Regarding the analysis section, an analysis of the differences in acoustic emission damage between no water pressure and stable water pressure has been added.

---

## [Decision Letter · Decision Letter 1]

8 Jul 2024

Tunnel and underground engineering rock mass water inrush damage and acoustic emission characteristics

PONE-D-24-14746R1

Dear Dr. Zeng,

We’re pleased to inform you that your manuscript has been judged scientifically suitable for publication and will be formally accepted for publication once it meets all outstanding technical requirements.

Kind regards,

Fabio Trippetta, Ph.D.

Academic Editor

PLOS ONE

Reviewers' comments:

Reviewer's Responses to Questions

**Comments to the Author**

1. If the authors have adequately addressed your comments raised in a previous round of review and you feel that this manuscript is now acceptable for publication, you may indicate that here to bypass the “Comments to the Author” section, enter your conflict of interest statement in the “Confidential to Editor” section, and submit your "Accept" recommendation.

Reviewer #1: All comments have been addressed

Reviewer #2: All comments have been addressed

2. Is the manuscript technically sound, and do the data support the conclusions?

Reviewer #1: Yes

Reviewer #2: Yes

3. Has the statistical analysis been performed appropriately and rigorously? 

Reviewer #1: Yes

Reviewer #2: Yes

4. Have the authors made all data underlying the findings in their manuscript fully available?

Reviewer #1: Yes

Reviewer #2: Yes

5. Is the manuscript presented in an intelligible fashion and written in standard English?

Reviewer #1: Yes

Reviewer #2: Yes

6. Review Comments to the Author

Reviewer #1: All my comments have been properly replied or addressed. I think it can be accepted in the present form.

Reviewer #2: The manuscript has been carefully revised and its quality has significantly improved, making it suitable for acceptance.

7. PLOS authors have the option to publish the peer review history of their article (what does this mean?). If published, this will include your full peer review and any attached files.

Reviewer #1: No

Reviewer #2: No

---

## [Editor Report · Acceptance letter]

11 Jul 2024

PONE-D-24-14746R1 

PLOS ONE

Dear Dr. Zeng, 

I'm pleased to inform you that your manuscript has been deemed suitable for publication in PLOS ONE. Congratulations! Your manuscript is now being handed over to our production team.

Kind regards, 

on behalf of

Prof. Fabio Trippetta 

Academic Editor

PLOS ONE